# Transplacental Transfer of Oxytocin and Its Impact on Neonatal Cord Blood and In Vitro Retinal Cell Activity

**DOI:** 10.3390/cells13201735

**Published:** 2024-10-19

**Authors:** Claudette O. Adegboro, Wenxiang Luo, Meha Kabra, Ryan M. McAdams, Nathaniel W. York, Ruwandi I. Wijenayake, Kiana M. Suchla, De-Ann M. Pillers, Bikash R. Pattnaik

**Affiliations:** 1Department of Pediatrics, Division of Neonatology and Newborn Nursery, University of Wisconsin–Madison, 1300 University Avenue, SMI 112, Madison, WI 53706, USA; cadegboro@pediatrics.wisc.edu (C.O.A.); mkabra2@wisc.edu (M.K.); mcadams@pediatrics.wisc.edu (R.M.M.); nwyork@wustl.edu (N.W.Y.); wijenayake@wisc.edu (R.I.W.); suchla@wisc.edu (K.M.S.); 2Department of Pediatrics, Division of Neonatology, University of Illinois at Chicago, Chicago, IL 60612, USA; wluo@uic.edu (W.L.); pillersd@uic.edu (D.-A.M.P.); 3McPherson Eye Research Institute, University of Wisconsin–Madison, 1300 University Avenue, SMI 112, Madison, WI 53706, USA; 4Endocrine and Reproductive Physiology Program, University of Wisconsin–Madison, 1300 University Avenue, SMI 112, Madison, WI 53706, USA; 5Children’s Hospital University of Illinois, University of Illinois Hospital & Health Sciences System, Chicago, IL 60612, USA; 6Department of Ophthalmology and Visual Sciences, University of Wisconsin–Madison, 1300 University Avenue, SMI 112, Madison, WI 53706, USA

**Keywords:** preterm, retinal vascularization, retinopathy, transcriptomic, connectivity, OXT

## Abstract

The development of fetal organs can be impacted by systemic changes in maternal circulation, with the placenta playing a pivotal role in maintaining pregnancy homeostasis and nutrient exchange. In clinical obstetrics, oxytocin (OXT) is commonly used to induce labor. To explore the potential role of OXT in the placental homeostasis of OXT, we compared OXT levels in neonatal cord blood among neonates (23–42 weeks gestation) whose mothers either received prenatal OXT or experienced spontaneous labor. Our previous research revealed that the oxytocin receptor (OXTR), essential in forming the blood–retina barrier, is expressed in the retinal pigment epithelium (RPE). We hypothesized that perinatal OXT administration might influence the development of the neural retina and its vasculature, offering therapeutic potential for retinal diseases such as retinopathy of prematurity (ROP). Plasma OXT levels were measured using a commercial OXT ELISA kit. Human fetal RPE (hfRPE) cells treated with OXT (10 µM) were assessed for gene expression via RNA sequencing, revealing 14 downregulated and 32 upregulated genes. To validate these differentially expressed genes (DEGs), hfRPE cells were exposed to OXT (0.01, 0.1, 1, or 10 µM) for 12 h, followed by RNA analysis via real-time PCR. Functional, enrichment, and network analyses (Gene Ontology term, FunRich, Cytoscape) were performed to predict the affected pathways. This translational study suggests that OXT likely crosses the placenta, altering fetal OXT concentrations. RNA sequencing identified 46 DEGs involved in vital metabolic and signaling pathways and critical cellular components. Our results indicate that the perinatal administration of OXT may affect neural retina and retinal vessel development, making OXT a potential therapeutic option for developmental eye diseases, including ROP.

## 1. Introduction

The placenta is a temporary fetal organ that attaches to the uterine lining during pregnancy, providing oxygen and nutrients to the developing fetus. Disruptions in placental homeostasis can significantly affect maternal and fetal health, highlighting the placenta’s vital role in a healthy pregnancy. Oxytocin (OXT), a neuropeptide produced by the hypothalamus and released into the bloodstream by the posterior pituitary gland, is well known for inducing uterine contractions during labor and milk letdown during lactation. However, OXT is a multifaceted regulator that maintains the placental microenvironment, optimizes placental function throughout pregnancy, and promotes fetal development. Given OXT’s key role in several placental functions, it is important to determine whether OXT crosses the placenta during gestation [1]. Nearly 30 years ago, Malek et al. first noted that OXT could cross the human placenta by simple diffusion. They determined that OXT crosses the placenta in both directions by simple diffusion, favoring the maternal-to-fetal direction [2]. However, it remains unclear whether the fetus produces endogenous OXT independently [3]. Understanding the sources of OXT during pregnancy is essential because it is not only a signal for parturition but also impacts overall fetal growth and development [4], vascular proliferation, and organogenesis [5]. Notably, there is controversy surrounding OXT’s ability to cross the placenta and affect fetal development. Our study investigated OXT’s transfer across the placenta throughout gestation, aiming to clarify the existing literature and provide insights into its role in fetal development.

There is a reasonable correlation between human eye development and the morphological stages of fetal development throughout gestation [6]. OXT has been detected in both human and animal fetuses as early as 16 weeks of gestation and coincides with the formation of retinal and retinal vessels during eye development [6]. The sharp increase in OXT levels, which occurs in the last trimester of pregnancy, is synchronous with rapid retinal development and vascular growth. This relationship suggests that OXT may play a role in regulating retinal and retinal angiogenesis [1]. The development of the retina begins at 28 days of gestation, and the retina is completely vascularized by 40 weeks of gestation [2]. In preterm infants, the abnormal development and vascularization of the retina can occur, leading to retinopathy of prematurity (ROP), a significant cause of blindness in this population. ROP is a disease caused by a combination of factors, including the cessation of normal retinal vascular growth and the suppression of growth-promoting factors following premature birth. Smith et al. proposed that deficiencies in developmental hormones and vascular growth factors may contribute to retinopathy pathogenesis [7]. Therefore, understanding the molecular action of OXT may provide a biomarker for retinal disorders or provide treatment options to prevent ROP.

In 1983, Gauquelin et al. first identified OXT in the human retina [7]. Despite these landmark findings, the literature rarely addresses the role of OXT in healthy neural retinal development [5]. The retinal pigmented epithelium (RPE) is a vital component of the neural retina and is particularly affected by ROP. The RPE is a monolayer of polarized pigmented epithelial cells interposed between the neural retina and choriocapillaris, performing vital functions to support the neural retina and maintain normal and healthy vision [8,9], such as forming a blood–retinal barrier, absorbing stray light, transporting nutrients and metabolites, performing retinoid recycling, and phagocytosing shed outer segments of photoreceptors (PRs). Although the RPE performs an essential function in healthy vision, and OXT regulates various critical physiological processes, including neurophysiology (like the maternal–fetal barrier), it is unknown whether OXT crosses the blood–retina barrier and alters physiology during health and disease. Additionally, the specific role of OXT in fetal retinal development remains unresolved, leaving a significant gap in understanding OXT’s impact on the retina during early life.

We previously identified oxytocin receptor (OXTR) mRNA and protein in the RPE of human retinas and rhesus monkey retinas, and in cultured hfRPE cells [10]. Additionally, we detected OXT in the retina of the rhesus monkey. These findings suggest that OXT plays a crucial role in the signaling mechanisms regulating retinal vascular development and may influence the abnormal retinal vascular growth often observed in ROP.

Our current study is based on two main hypotheses. First, maternal OXT crosses the placenta and alters fetal OXT concentrations. Second, when mimicking fetal circulation, systemic OXT within the fetal circulation affects the RPE transcriptome in cultured hfRPE cells, indicating its potential role in retinal development and visual function regulation.

## 2. Methods

### 2.1. Patient Recruitment

The clinical portion of this study was designed as a prospective cohort study. Preterm and term neonates were recruited from our birthing center, UnityPoint Health (UPH) Meriter Hospital, a Level III NICU in Madison, Wisconsin. The UPH Meriter Hospital Institutional Review Board approved the study. The recruitment period occurred over 18 months, from October 2017 to March 2019. Neonates born at gestational ages ranging from 23 to 42 weeks via cesarean section or vaginal delivery were included. Infants were included randomly, depending on the ability of the nursing staff to collect cord blood. Neonates were excluded from the study if they were born at an outside hospital, had an umbilical cord anomaly, or had insufficient cord blood volume. The following information was obtained from each study participant: gestational age, sex, birth weight, presence of maternal illness, synthetic OXT administration to augment labor, intrapartum steroid administration, and need for oxygen during delivery resuscitation.

### 2.2. Consent Process

As per UnityPoint Health—Meriter’s IRB requirements, verbal consent was obtained via nursing staff for this study. The study involved the collection of cord blood from discarded umbilical cord segments, which would otherwise have been routinely discarded unless a plan for special testing was in place. Given that the samples were collected from discarded biological materials, the IRB determined that written consent was unnecessary. The verbal consent process, approved by the IRB, ensured that parents were informed about the study and the use of the discarded cord blood. The IRB deemed this sufficient due to the minimal sample collection risk. All cord blood samples were stored and frozen for later analysis, and all demographic and clinical data were securely stored, de-identified, managed, and analyzed solely by researchers.

### 2.3. Oxytocin Extraction

Our group conducted a pilot study (unpublished), which determined that OXT plasma levels begin to decrease 5 days after delivery. With this knowledge, blood samples for OXT assays in the current study were collected from mixed umbilical artery and vein samples at birth. This approach was chosen due to the practical difficulties of obtaining separate arterial and venous samples. It should be noted that umbilical arteries, which are only 1–4 mm in diameter, are often challenging to sample individually for blood collection. Samples for cord blood testing samples are often mixed venous–arterial. Based on this, we determined that mixed samples should provide a sufficient representation of fetal OXT levels while allowing us to collect the necessary data consistently across all participants.

Samples were then placed into a prechilled 1.0 mL purple top ethylenediaminetetraacetic acid (EDTA) microcontainer tube and refrigerated for up to four days. The samples were centrifuged at 1000× *g* for 15 min at 4 °C to obtain plasma, which was subsequently stored in aliquots at −80 °C. OXT immunoreactivity levels were quantified in duplicate via a commercial OXT ELISA kit (Enzo Life Sciences, Inc., Farmingdale, NY, USA), as noted in a similar study [11]. In this study, the assayed OXT samples had a 15.0 pg/mL sensitivity and inter- and intra-assay coefficients of variation less than 20.9%.

Notably, high-quality plasma OXT extraction is difficult and often results in a significant loss of measurable OXT [12,13]. This study used a commercially available OXT ELISA kit from Enzo Life, validated for cross-reactivity, accuracy, precision, and recovery. Given the labile nature of this critical gestational hormone, we followed a strict protocol (Thermo Fisher Scientific, Waltham, MA, USA, 2007) to limit the rapid enzymatic breakdown of OXT.

Sample extraction was evaluated via two different methods: solid-phase extraction and solvent extraction. Solid-phase extraction was performed using 200 mg C18 Sep-Pak columns (Bachem, San Carlos, CA, USA). The columns were equilibrated twice with 3 mL of acetonitrile and then with 3 mL of 0.1% trifluoroacetic acid (TFA). Up to 1 mL of plasma was mixed with an equal volume of 0.1% TFA, the mixture was centrifuged at 17,000× *g* for 20 min at 4 °C, and then the acidified supernatant (plasma) was added to the column. OXT was eluted with 3 mL of 60% acetonitrile. The solvent was evaporated to dryness via a speed vacuum centrifugal concentrator.

Solvent extraction involved preparing samples with ELISA reagents and extrapolating the data on a microplate reader. Fetal OXT concentrations were recorded and plotted against a 4-parameter logistic (4PL) regression curve. All the measurements were performed to improve accuracy. We analyzed the data via MyAssays Analysis Software Solutions and Inkscape (version 0.92.3).

### 2.4. Data Analysis

We performed a quantitative analysis of the samples via a 4PL curve. All samples were compared to a set of known standard quantities to determine the amount of OXT in each sample. This data comparison generated a predicted standard curve and produced an ideal one. Once the standard curve was generated, the location of the sample was determined by the fit curve, and values were interpolated. We used a quadratic equation to fit the model data and produce a sigmoidal curve.

### 2.5. Statistical Analysis

Maternal and neonatal characteristics were summarized using medians and interquartile ranges (IQRs; range between the 25th and 75th percentiles) for continuous characteristics or with frequencies and percentages for categorical factors. Comparisons between groups (preterm neonates versus term neonates) were made via nonparametric rank-sum tests, chi-square tests of associations, or Fisher’s exact tests in instances where the expected frequency was less than five. Multiple linear regression was used to address whether the cord blood concentration of OXT was associated with one or more maternal or infant characteristics. Statistical significance was prespecified to 0.05, and no adjustment for multiple testing was made in this exploratory research.

Additionally, the cord blood concentration was log-transformed before analysis to improve symmetry. The means calculated for log-transformed data represent geometric means upon back-transformation and serve as an estimator of the median response on the original scale. At the same time, differences between means on the log scale become multiplicative effects (i.e., fold changes) involving the estimated medians after back-transformation to original units [14].

### 2.6. Human Fetal RPE Cell Cultures

We obtained approval from the institutional review board at the University of Wisconsin–Madison to use commercial human fetal cell lines. Passage 2 cryopreserved primary Clonetics human RPE cells (hfRPE) (LONZA, Walkersville, WA, USA) were cultured in 75 cm^2^ flasks in an admixture of hfRPE culture media (MEM alpha base medium [Gibco, Grand Island, NY, USA]), N1 supplement, glutamine (Gibco), pen-strep (Gibco), MEM nonessential amino acids, taurine, hydrocortisone, and 3,3′,5-triiodothryonin + 10% fetal bovine serum (FBS) (Gibco) for 48 h. At 70% confluence, the cells were exposed to 1x EDTA-trypsin (LONZA) for 4 min at 37 °C in 5% CO_2_. The cells were then collected in hfRPE culture media supplemented with 8% FBS. The cells were subsequently seeded at approximately 1 × 104 cells/cm^2^ density onto 24-well culture plates. Finally, the cells were cultured in hfRPE media + 8% FBS until they reached 95% confluence. At this point, the cells were maintained in hfRPE media + 0% FBS, and the media was changed every 2 days. Human fetal RPE cells cultured in plates for 4 to 5 weeks were treated with or without 10 μM OXT (Sigma-Aldrich, St. Louis, MO, USA) for 12 h, and this experiment was repeated four times. The treated RPE cells were further assayed for gene expression.

### 2.7. RNA Sequencing and Data Analysis

RNA was extracted from cultured RPE cells via an RNeasy Mini Kit (Qiagen, Hilden, Germany) and treated with DNase according to the manufacturer’s instructions. The concentration of RNA was measured via a NanoDrop spectrophotometer (Thermo Fisher Scientific, Waltham, MA, USA), and the integrity of the RNA was assessed via an Agilent 2100 Bioanalyzer (Agilent Technologies, Santa Clara, CA, USA). RNA sequencing was performed at the Gene Expression Center of the Biotechnology Center at the University of Wisconsin–Madison. The transcriptome libraries were sequenced on an Illumina HiSeq2500 platform in 1 × 100 base-pair mode. The sequence data were analyzed via the University of Wisconsin–Madison Bioinformatics Resource Center with the RNAseq pipeline v1.0. In this RNA-seq pipeline v1.0, the trimming software skewer removed adapters in the base call raw reads [15]. Genes with zero or low-abundance counts were filtered [16]. Low-abundance genes were defined as those with an average read count below a threshold of 1.0 in two or more samples. The sequence read counts for the experimental samples were normalized to the trimmed mean of the M values (TMM) [17]. The trimmed sequence reads were aligned against the hsapiens_gene_ensembl human genes’ (GRCh38.p10) reference genome sequence via STAR (Spliced Transcripts Alignment to a reference) [18]. Aligned and mapped reads for genes and transcripts were counted in each sample via RSEM (RNA-seq by expectation maximization) [19]. Differentially expressed genes (DEGs) were analyzed with General Linear Models (GLM) via the edger package [20]. The change in gene expression between the two groups was reported on a base 2-logarithmic scale of fold change: log2-fold change (FC). A gene was defined as differentially expressed when the adjusted *p*-value and false discovery rate (FDR) was ≤0.05, the FC between the treatment and control groups was ≥1.5, or the log2 (FC) was ≥0.585. Benjamini–Hochberg correction was applied to control the FDR [21]. We identified 14 downregulated and 32 upregulated genes based on these thresholds.

Real-time PCR was carried out to validate the RNA transcriptomics findings. Some of the genes found to be upregulated, downregulated, and involved in cholesterol biosynthesis were validated independently via specific primer pairs via real-time PCR (Appendix A). For this purpose, hfRPE cells were seeded in a 24-well plate and cultured for 4 to 5 weeks. Mature hfRPE cells were treated with various concentrations of OXT (0.01, 0.1, 1, or 10 µM) for 12 h. Treated RPE cells were collected, and RNA was isolated via an RNeasy Plus Mini Kit (Qiagen). For gene expression analysis, RNA was converted to cDNA via a high-capacity cDNA reverse transcription kit (Applied Biosystems, Waltham, MA, USA), and real-time PCR was performed via SYBR green PCR master mix on a QuantStudio 3 (Applied Biosystems). The *GAPDH* gene was used as an endogenous control in the relative quantification (∆∆Ct) method. The average Ct values of each target gene were normalized to the respective average Ct values of *GAPDH* to obtain the ∆Ct value (∆Ct = average Ct of the target gene − average Ct of GAPDH). The ∆∆Ct value was calculated as ∆Ct _treated_ − ∆Ct _untreated,_ and the fold change was calculated via the 2^−∆∆Ct method^. The untreated cells were used as a reference. The experiment was repeated three times. Student’s *t*-test (paired two-tailed) was used to compare two groups, and ANOVA was used to compare multiple groups. The significance threshold (*p*-value) was set at 0.05 and 0.001.

### 2.8. Functional Analysis of Differentially Expressed Genes

Gene Ontology (GO) term and pathway enrichment analyses were conducted for the DEGs via the g:Profiler platform [22]. GO covers three domains: molecular function, biological process, and cellular component. The Kyoto Encyclopedia of Genes and Genomes (KEGG) database was used for pathway enrichment analysis. FDR controlled by the Benjamini–Hochberg procedure was used in multiple testing correction enrichment analyses. An FDR ≤ 0.05 was considered to indicate significant enrichment. In addition, DEGs satisfying the FDR (≤0.05) and Log2FC (>1 or >−1) thresholds were subjected to enrichment analysis via the functional enrichment analysis tool (FunRich 3.1.3.exe) accessed on 11 September 2023 (http://funrich.org/index.html). The list of genes was uploaded as two different datasets (upregulated and downregulated) for functional enrichment and compared under six categories: clinical phenotype, cellular component, molecular function, biological process, biological pathway, and site of expression.

A network analysis was performed via the Cytoscape (version 3) plug-in GeneMANIA, a web interface for analyzing gene or protein lists and predicting gene function based on datasets collected from publicly available databases. These datasets include coexpression and colocalization data from the Gene Expression Omnibus (GEO), physical interaction data from BioGRID, predicated protein interaction data from Orthology I2D, and a consolidated protein interaction database (IRefIndex) [23]. In analyzing a long (five or more) query list of genes or proteins, GeneMANIA will likely identify more genes or proteins similar to those noted in the list above. In addition, it will identify physical or predicated physical interactions between proteins, coexpression, and networks of interactions among query genes or proteins [23]. We used the default GeneMANIA parameters and removed additional genes similar to those identified by GeneMANIA to find a network among the DEGs identified in our RNAseq analysis.

## 3. Results

### 3.1. Study Characteristics

The in vivo portion of our study included 74 infants, 54 (73%) of whom delivered at term, and Figure 1 depicts the study design. The average gestational age was 39 weeks, and the average birth weight was 3.4 kg. Figure 2 shows the results of the recruitment process. One hundred twenty neonatal samples (Table 1) were collected over ten months; ten had missing data. We used samples that had complete information for OXT extraction. However, 36 samples did not yield an OXT measurement. As noted previously, because OXT is a volatile substance, obtaining high-quality measurable extracts has generally proven difficult for researchers. A total of 74 samples had measurable OXT extracted. Compared with term deliveries, preterm births were associated with more frequent labor, high blood pressure (HBP)/pre-eclampsia, diabetes, and maternal steroid use. There were no cases of chorioamnionitis. Placental abruptions or 5 min Apgar scores less than 5 were observed in only one or two cases. OXT was administered to mothers in 45% of the preterm deliveries and 30% of the term deliveries (*p* = 0.214). We did not find any associations between cord blood concentrations of OXT and gestational age (Figure 2, left panel) or birth weight (Figure 2, right panel; rs = −0.17, *p* = 0.16). There was also no correlation between cord blood OXT concentration and preterm versus term infants (*p* = 0.106; Table 1).

### 3.2. Maternal Oxytocin Administration Crosses the Placenta

Among preterm infants, maternal exposure to OXT before delivery resulted in a median cord blood OXT concentration 3.9 times greater (95% CI: 2.0–7.7, *p* < 0.001) than that of infants whose mothers did not receive OXT. A similar effect was observed in full-term infants, with a median fold change of 2.8 (95% CI: 1.7–4.4, *p* < 0.001). These effects, however, are not significantly different (*p* = 0.42) and suggest that a modest increase in fetal OXT levels is associated with maternal OXT administration (Figure 3). After controlling for gestational age, the median cord blood OXT concentration among infants whose mothers received OXT was 3.1 times greater (95% CI: 2.1–4.5, *p* < 0.001) than that among infants whose mothers did not receive OXT (186.2 pg/mL vs. 60.58 pg/mL).

### 3.3. Maternal Labor Affects Fetal OXT Concentrations

As shown in Figure 3, maternal OXT administration during labor resulted in significantly higher median cord blood OXT concentrations than OXT administration alone (*p* < 0.001). However, labor alone had no significant effect (*p* = 0.108) on OXT concentrations. When both labor and maternal OXT administration were considered, the median cord blood OXT concentration was 2.53 times greater (95% CI: 1.68–3.81) than when only labor was present and 3.50 times greater (95% CI: 2.30–5.31) than that in the groups without labor and OXT administration (Figure 3, *p* < 0.001).

### 3.4. Identification of Key Pathway-Associated Genes by GO and KEGG Term Enrichment

From a molecular standpoint, RNA sequencing revealed a total of 14,975 genes (unfiltered data), including upregulated (n = 301), downregulated (n = 123), and nonsignificant (n = 14,551) genes, in OXT-treated RPE cells compared with those in control cells. The data were filtered out based on the Log2FC (<0.585) and FDR (≤0.05) thresholds, and 109 DEGs were subcategorized into upregulated (n = 78) and downregulated genes (n = 30). (Figure 4, Appendix A). GO and KEGG term enrichment analyses were performed to understand the biological mechanisms of these DEGs in treated RPE cells. In the analysis of the upregulated DEGs, the enriched KEGG terms included metabolic pathways (KEGG:01100), steroid biosynthesis (KEGG:00100), terpenoid backbone biosynthesis (KEGG:00900), the PI3K-Akt signaling pathway (KEGG:04151), the calcium signaling pathway (KEGG:04020), the cAMP signaling pathway (KEGG:04024), the regulation of the actin cytoskeleton (KEGG:04810), the PPAR signaling pathway (KEGG03320), the ECM–receptor interaction (KEGG:04512), focal adhesion (KEGG:04510), and the phagosome (KEGG:04145) (Figure 5; Appendix A). The enriched GO terms for the upregulated DEGs included membrane (GO:0016020), extracellular matrix (GO:0031012), focal adhesion (GO:0005925), cytoskeleton (GO:0005856), protein binding (GO:0005515), ion binding (GO:0043167), signaling receptor binding (GO:0005102), catalytic activity (GO:0003824), calcium ion binding (GO:0005509), extracellular matrix binding (GO:0050840), low-density lipoprotein particle receptor activity (GO:0005041), metabolic process (GO:0008152), signal transduction (GO:0007165), cholesterol biosynthetic process (GO:0006695), cellular component organization or biogenesis (GO:0017840), and cell adhesion (GO:0007155) (Figure 5; Appendix A). The enriched KEGG terms of the downregulated DEGs included prostate cancer (KEGG:05215), EGFR tyrosine kinase inhibitor resistance (KEGG:01521), and the MAPK signaling pathway (KEGG:04010) (Appendix A). The enriched GO terms for the downregulated DEGs included regulation of epithelial cell proliferation (GO:0050678) and cell migration (GO:0016477) (Appendix A).

FunRich analysis of 14 downregulated and 32 upregulated genes (FDR: ≤0.05, Log2FC: >1 or >−1) revealed an enrichment analysis of the top 15 characteristics across the six categories (Figure 5). In the clinical phenotypes category, the most upregulated genes were associated with conjunctivitis, cataracts, corneal abrasions and scarring, eyelid ulcerations, and neoplasia. On the other hand, downregulated genes were found to be involved in microphthalmia, anophthalmia, postnatal growth retardation, and cardiovascular and neurological diseases. Some up- (33%) and downregulated genes (50%) were involved in common phenotypes related to hair, eyes, and growth. The upregulated genes in the mitochondrion, Golgi apparatus, integral to membrane, cytoskeleton, and cytoplasmic vesicle categories were enriched, whereas the downregulated genes in the endosome, extracellular matrix (ECM), and endoplasmic reticulum categories were enriched. A percentage of the up- and downregulated genes were also observed in the plasma membrane, nucleus, cytoplasm, exosomes, lysosomes, extracellular space, and nucleolus. Among the upregulated genes, genes related to molecular activity included chaperons, transaminases, transcription regulators, growth factors, G-protein coupled receptors, auxiliary transport proteins, and receptor signaling. Ion channels, transcription factors, ECM constitution, transporters, and catalytic activity were observed solely in the downregulated genes. Some molecular functions, such as calcium ion binding and metallopeptidase activity, were common among the up- and downregulated genes. Unlike downregulated genes, upregulated genes are involved in biological processes such as regulating exocytosis, hormone secretion, anti-apoptosis, and the immune response. However, genes in the up- and downregulated categories belonged to biological processes such as metabolism, cell communication, signal transduction, energy pathways, cell growth and maintenance, regulation of nucleic acid metabolism, and transport. The upregulated genes were associated with biological pathways such as granulocyte–macrophage colony-stimulating factor (GMCSF)-mediated signaling, vascular endothelial growth factor (VEGF) signaling, signal transduction, integrin–cell surface interactions, peptide ligand-binding receptors, the immune system, transmission across synapses, and G-protein coupled receptor (GPCR) signaling. Downregulated genes were associated with pathways such as smooth muscle contraction, transmembrane transport of small molecules, biological oxidations, multifunctional anion exchangers, and axon guidance. Both sets of genes were associated with metabolism and developmental pathways. The expression sites of the up- and downregulated genes included the kidney, plasma, skeletal muscles, brain, skin, heart, tears, dendritic cells, lacrimal glands, and embryonic stem cells. Unlike those of the downregulated genes, the expression sites of the upregulated genes included the cornea, eyes, macrophage lens, and blood vessels.

### 3.5. Protein Function Network Identified by the Cytoscape Plug-in GeneMANIA

We further analyzed interactions among 14 downregulated and 32 upregulated genes (FDR: ≤0.05, Log2FC: >1 or >−1) (Appendix A) via the Cytoscape (version 3) plug-in GeneMANIA. This analysis identified these interactions and constructed an interaction network among genes from our query list and the genes predicted by GeneMANIA (Figure 6). We removed interactions among our list of genes with predicted genes to strengthen the network among our 46 genes of interest. The genes in our query lists were in a network based on various interactions, including coexpression, colocalization, and physical and predicted interactions. The resulting network revealed that coexpression accounted for 80.07%, physical interactions accounted for 9.51%, and colocalization accounted for 7.92% of the total network weights. Moreover, 2.49% of the interactions were predicted. Thus, coexpression was among the most common interactions shown in Figure 6.

Complex networking was observed among the up- and downregulated genes. A cluster of colocalization was formed between *SLC14A1* and *SMCO1*, *PDGFB* and *TGM2*, and among *PTPRB*, *ESM1*, *TMEM100*, *HBEGF*, *ITGA2*, and *MALL*. Our query list’s genes with physical interactions included *IL7R* and *MALL*, *ITGA2* and *MMP1*, *ANKRD1*, *ESM1*, *TRIM55*, and *DAPL1*. A predicted interaction was observed between only *ITGA2* and *MMP1*. A few genes (*CLEC18C*, *CACNA1E*, *COL21A1*, *ARL2-SNX15*, *C11orf88*, and *NLRP10*) did not interact with other genes in our query list. The remaining genes from our query list were found to be in complex networks based on their coexpression (Figure 6).

### 3.6. OXT Treatment of hfRPE Cells Induces Dynamic Modifications in the Expression Profile of Genes

Whose expression is upregulated or downregulated, as identified via RNA sequencing, and the expression profiles of genes involved in cholesterol biosynthesis and metabolism pathways were further validated via real-time PCR. We selected four downregulated genes, seven upregulated genes, and thirteen genes previously reported to be involved in cholesterol biosynthesis (PMID: 35275950) to study their expression profiles in OXT-treated hfRPE cells. Our findings confirmed the significant downregulation of two genes (*RNVU1–7*, *p* < 0.05 and *FMO1*, *p* < 0.01) in 1, 0.1, and 0.01 µM OXT-treated HfRPE cells, whereas two of the genes remained undetected (*ANPEP* and *SLC14A1*) in these cells (Figure 7A). These data support and align with the findings of RNA sequencing. In contrast to our RNA sequencing findings, only the *MMP1* gene (*p* < 0.05) was significantly upregulated, whereas two genes (*XIRP1* and *KISS1*) were downregulated (*p* < 0.05). A few of the genes (*ESM1*, *IL7R*, and *CLEC18C*) that were found to be upregulated by RNA sequencing were not detected by real-time PCR (Figure 7B). The *GABBR2* gene presented different expression profiles in response to multiple doses of OXT. Significant downregulation (*p* < 0.05) was observed with a higher dose (1 µM), whereas upregulation (*p* < 0.05) was observed with lower doses (0.1 and 0.01 µM) of OXT (Figure 7B) (Appendix A).

We selected a list of genes involved in cholesterol biosynthesis and metabolism to study their OXT-induced expression modulation in HfRPE cells. We observed that, except for two genes (*CYP46A1* and *LDLR*), the expression of most of these genes was significantly altered with almost all the doses of OXT. Significant downregulation (*p* < 0.05 or *p* < 0.01) of nearly all the genes (*SRBII, ABCA1*, *ABCA4*, *RPE65*, *HMGCR*, *SREBP1*, *SREBP2*, *CYP11A1*, and *CYP27A1*) was observed, whereas *APOE* was not detected in our treated or untreated hfRPE cells (Figure 7C) (Appendix A).

## 4. Discussion

This translational study suggests that OXT is likely to cross the placenta and affects fetal OXT concentrations. Specifically, maternal exposure to OXT resulted in infant cord blood OXT levels being 2–4 times higher than those in infants whose mothers were not exposed to synthetic OXT. These findings were statistically significant and strongly associated with the presence or absence of labor. RNA sequencing revealed that 46 genes were differentially expressed in OXT-treated RPE cells. These up and downregulated genes were related to vital metabolic and signaling pathways and critical cellular components.

Additionally, a complex cluster of coexpressed and predicted interaction networks was identified based on our query list of DEGs and the list of genes predicted by GeneMANIA, indicating that OXT also influences the RPE transcriptome. Many colocalizations occurred among the proteins of these genes as well. Our clinical and scientific findings suggest that OXT crosses the placenta and may have the ability to increase fetal OXT concentrations in addition to regulating retinal and retinal vascular growth. These findings indicate that either OXT that is crossing or another entity that OXT induces may be a potential early therapeutic target for ROP.

Notably, RPE cells undergo terminal differentiation early in development. However, they remain dormant and have little or no cell turnover throughout normal life [24,25]. Thus, maintaining structural, metabolic, and functional homeostasis among RPE cells is vital for decoding visual signals. In the present study, we revealed that OXT-regulated, transcribed genes function in metabolic pathways, cholesterol biosynthetic processes, phagosomes, focal adhesions, actin cytoskeleton regulation, PPAR signaling pathways, PI3K–Akt signaling pathways, extracellular matrices, and ECM–receptor interactions. These regulated pathways involve multiple cellular functions, including cell membrane phagocytosis, cell survival and protection, and the structural integrity of the ECM in RPE cells.

This study used multiple GO and KEGG terms related to cholesterol biosynthesis to analyze the DEGs upregulated by OXT. Cholesterol is an essential constituent of the cell membrane. It regulates various cellular processes, including membrane trafficking, ligand binding, receptor recycling, and signal transduction [26,27]. Maintaining cholesterol homeostasis is necessary for normal cell function and viability. These GO and KEGG terms included metabolic pathways, steroid biosynthesis, terpenoid backbone biosynthesis, metabolic processes, and cholesterol biosynthetic processes. Further analysis of those genes via the GeneMANIA network revealed a close cluster of genes involved in cholesterol biosynthetic pathways and regulation. Among this cluster of genes, eight are cholesterol biosynthetic enzymes: one is a receptor for the endocytosis of cholesterol, and the other is a transcription factor that suppresses cholesterol biosynthetic enzymes [26,27,28,29,30,31]. Our quantitative validation of the genes implicated in the synthesis, uptake, elimination, and regulation of cholesterol levels revealed significant downregulation. Most importantly, *HMGCR*, a rate-limiting enzyme of cholesterol biosynthesis, and CYP27A1, which helps eliminate cholesterol from the retina, were downregulated at the mRNA level. Thus, our data suggest that OXT affects cholesterol biosynthesis and homeostasis in RPE cells by modulating the transcript expression of various genes. When cholesterol homeostasis in the RPE is impaired, vision is affected by macular degeneration [32]. Zheng et al. demonstrated the presence of HMGCR in human and murine RPE cells via immunohistochemical analyses [33,34]. Ramachandra et al. studied the human induced pluripotent stem cell (iPSC)-derived RPE and reported active cholesterol biosynthesis [35]. Biswas et al. found cholesterol in an immortalized human RPE-derived cell line (ARPE-19) via radiolabeled acetate [36].

We found that genes whose expression was upregulated by OXT treatment were involved in phagocytosis. These genes included those involved in focal adhesion, the regulation of the actin cytoskeleton, phagosome formation, and calcium signaling. To understand the relevance of these findings, we reviewed the essential and functional components of the human retina. When photoreceptors are exposed to light, damaged proteins and lipids generate photo-oxidative radicals [37,38]. These toxic substances accumulate in the photoreceptor each day [37,38]. Photoreceptors constantly remove these substances when their outer segments are renewed to maintain normal function [37,38]. In this renewal process, a new portion of the photoreceptor outer segment (POS) is built, and the POS tip containing a high concentration of toxic substances is shed from every photoreceptor [37,38]. A critical role of the RPE is phagocytosis of the shed POS to maintain the photoreceptor’s structural and functional homeostasis [8,37]. The phagocytosis of the POS by the RPE is a sequence of events that includes the binding of the POS to the RPE, ingestion of bound POS, maturation of the phagosome, enzymatic breakdown of macromolecules, and resolution of the RPE phagolysosome [8,37]. The α_v_β_5_ integrin receptors are involved in the binding of POSs to the RPE [8,37,39,40], focal adhesion kinase (FAK) is activated after binding, and phosphorylating receptor tyrosine kinase c-mer (MerTK) initiates subsequent events [8,37,39,40]. These events include the activation of intracellular inositol 1,4,5-trisphosphate (InsP3)/Ca^2+^ and the rearrangement of F-actin, leading to bound POS ingestion [37,41,42,43].

Calcium signaling mediates phagosome maturation, which requires the fusion of phagosomes with endosomes and lysosomes [41]. FAK is the key kinase in focal adhesion. It mediates the POS ingestion process by the regulation and rearrangement of actin [37,39,44]. Phagosomes (cellular compartments) are needed to remove toxic radicals and macromolecules from POSs [37,45]. Furthermore, RPE phagocytosis has circadian rhythms [46]. Calcium signaling mediates phagosome maturation and may be the driving force for POS-shedding rhythms [46]. Interestingly, OXT was found to regulate Tetrahymena phagocytosis [47].

This study revealed multiple enriched pathway terms with a protective function in the RPE from OXT-upregulated DEGs. These terms included the PI3K-Akt signaling pathway, the PPAR signaling pathway, and the cAMP signaling pathway. Previous studies have demonstrated that the PI3K-Akt and PPAR signaling pathways each mediate the protective actions of various factors and pharmacological inhibitors of oxidative stress in RPE cells, ultimately promoting cell survival [48,49,50,51,52]. Notably, RPE cells undergo constant oxidative stress from multiple sources [53,54,55]. The RPE is located close to the choriocapillaris and receives high blood flow rates that are well saturated with oxygen [53,54]. Long-term light exposure increases the interaction of light with endogenous chromophores in the RPE, causing excitation that results in a highly reactive state [53,54]. Highly reactive chromophores rapidly interact with other molecules, including oxygen, generating reactive oxygen species (ROS) [53,54]. Furthermore, a robust amount of metabolic activity occurs in the mitochondria of the RPE to meet its high energy needs, which also generates a high concentration of ROS [53,54]. These various sources of oxidative stress can elicit harmful effects on biomolecules and the RPE [53,54,55] mitochondrial network. Thus, it is essential to protect the RPE from oxidative damage to preserve its integrity and function. In addition, mature human RPE cells undergo minimal proliferation and have a limited proliferative response to injury. Pathologic processes generally accompany marked proliferation in the human RPE [24,25]. For this reason, controlling the proliferation of RPE cells is essential. Hecquet et al. reported that the cAMP signaling pathway inhibits RPE proliferation via the mitogen-activated protein kinase pathway [56]. Furthermore, the cAMP signaling pathway also plays a role in transducing the protective function of klotho, an aging-suppressor gene, in RPE cells [57]. 

Our study identified enriched GO and KEGG terms related to the extracellular matrix (ECM) from the upregulated genes. These terms included ECM–receptor interaction, extracellular matrix, and extracellular matrix binding. One of the critical functions of the RPE is transporting nutrients and metabolic waste products from photoreceptors across Bruch’s membrane (BrM). BrM houses RPE cells and is located between the RPE and choriocapillaris [8,9]. Thus, it is essential to maintain both a stable interaction between the RPE and BrM and a stable or appropriate composition within BrM [8,9,58]. Otherwise, alterations in these properties can lead to ocular pathology [9,58]. RPE cells produce ECM components within BrM and express receptors that bind to the ECM [59]. Sugino et al. demonstrated that cell-deposited matrices promote RPE survival in aged human BrM specimens [60]. Several aspects of this fundamental study demonstrate that these findings suggest that OXT may modulate BrM and the RPE, and the interactions between them, potentially influencing RPE function and retinal health.

Among those downregulated genes, the enriched terms included prostate cancer, EGFR tyrosine kinase inhibitor resistance, and the MAPK signaling pathway. Activating the pathways included in these terms can lead to cellular proliferation, migration, and invasion. The MAPK signaling pathway is an essential pathway that mediates the proliferation of the RPE [56]. The downregulation of this pathway may have a protective effect on RPE cells [24,25]. OXT treatment activates, suppresses, or modulates various pathways that perform diverse functions. This study’s findings, like previous studies, indicate that OXT-regulated pathways mediate essential RPE functions, including cholesterol biosynthesis and homeostasis, phagocytosis, RPE protection, and Bruch’s membrane integrity (Figure 8).

For the clinical purpose of our study, we investigated whether OXT is transferred across the placenta. Our study included 74 near-term infants. Compared with term deliveries, preterm births were associated with more frequent episodes of spontaneous labor (HBP)/preeclampsia, diabetes, and antenatal steroid use. No significant associations were found between cord blood OXT concentrations and gestational age or birth weight. However, the administration of OXT during labor resulted in significantly higher median cord blood OXT concentrations than the absence of exogenous OXT. This finding suggests that OXT crosses the placenta and equilibrates between the mother and fetus. When both labor and OXT were present, the median cord blood OXT concentration was 2.53–3.50 times higher than when only one or none were present. Hence, it is likely that the administration of synthetic OXT during labor increases fetal OXT levels, regardless of gestational age. The presence of labor also showed a positive association with increased fetal OXT concentrations. This translational study represents a critical step in understanding the role of OXT in human gestation.

This aspect of our study is unprecedented: monitoring the transfer of OXT across the maternal–fetal barrier in vivo. Previous studies have shown that OXT crosses the placenta in in vitro models. Malek et al. investigated OXT and inulin transport across human placental tissues, reporting similar transfer rates in both directions in vitro. Their study concluded that OXT likely crosses the placenta via simple diffusion, with greater permeability in the maternal-to-fetal direction compared to fetal-to-maternal transport [2]. Our results demonstrated that fetal OXT concentrations were significantly greater among mothers who received OXT before delivery. This finding strongly implies that OXT crosses the maternal–fetal barrier during human gestation. 

Preterm and term neonates whose mothers underwent labor and received OXT before delivery had significantly higher OXT concentrations than those whose mothers did not undergo or receive OXT before delivery. This statistical significance was especially true among the preterm group, highlighting that many preterm neonates are likely born due to maternal preterm labor [61]. Furthermore, our findings imply that maternal OXT concentrations not only increase in the presence of labor or at birth but also likely cross the maternal–fetal barrier and contribute to surges in fetal plasma OXT levels. Additional findings included associations between preterm delivery and maternal illness. The frequency of deliveries involving diabetes and HBP/preeclampsia among preterm births was greater than those among term deliveries when maternal and infant birth characteristics were reviewed. Preterm births were also associated with the administration of antenatal steroids and the need for oxygen or advanced respiratory support during resuscitation.

The clinical arm of our study is the first to monitor OXT transfer across the maternal–fetal barrier in vivo and at several time points throughout human gestation (23–42 weeks). Research using in vitro models has shown that OXT can cross the placenta via simple diffusion [2]. Our study, conducted on the maternal–infant dyad in “real-time”, is consistent with these models and demonstrated significantly higher fetal OXT concentrations among mothers who received OXT before delivery (especially in the presence of labor). This is a significant strength of our study, and these findings strongly support the theory that maternal OXT not only crosses the placenta but also contributes to surges in fetal OXT levels.

Additionally, we aimed to achieve accurate and effective OXT extraction via an advanced protocol. While prior studies have only measured OXT via an in situ approach (to conserve blood volume), we successfully extracted OXT from whole blood volumes as small as 50 microliters [62,63]. This novel technique uses organic solvents and HPLC columns (solid-phase extraction) to increase the ability of ELISA kits to detect and extract highly volatile molecules such as OXT from their natural biological matrix (plasma).

Our study had several limitations. A primary limitation was the considerable variation in fetal OXT concentrations. Although our findings strongly suggest that OXT crosses the placenta, our novel extraction protocol in this pilot study was likely subject to procedural error and varying concentrations. A more efficient protocol is needed for future studies to ensure optimal OXT extraction. Another limitation was the collection of blood samples from mixed umbilical artery and vein sources. Due to the technical challenges of sampling umbilical arteries, which are only 1–4 mm in diameter, obtaining separate arterial and venous samples was not feasible. This may introduce variability, as arterial and venous blood differ in composition. However, mixed samples provided a reasonable representation of fetal OXT levels, and this approach is commonly used in similar studies. Future research could investigate the impact of separate arterial and venous samples to refine our understanding further.

Additionally, the small and unequal sample size between term and preterm infants was a limitation. Our statistical methods of log transformation helped improve the symmetry between the two cohorts and allowed for improved statistical power. However, our study was not initially powered with equal control arms or an ideal number of study participants to identify the transplacental effects of OXT. We also recognize that RNA-seq-based approaches, while widely accepted in the scientific community, have technological limitations that do not account for the contributions of noncoding sequences.

This translational study, however, shows promise for further investigation of OXT as a gestational hormone essential for the normal development of the retina. As OXT likely transfers from mother to fetus, it is possible that OXT concentrations continually increase throughout human gestation while the retina matures, surging just before birth and in the presence of labor. If this effect is valid, OXT could affect certain vascular growth factors expressed during ROP’s pathologic and vasoproliferative stages. Future studies will need larger cohorts with infants at varying gestational ages to assess the specific trajectory of OXT among the maternal–infant dyad throughout human gestation. It may also be worthwhile to examine whether delivery methods play a role in transferring OXT across the maternal–fetal barrier.

While our study provides valuable preliminary insights into the transplacental transfer of OXT and its association with fetal outcomes, its generalizability is limited. The findings most apply to populations similar to our study, as the small sample size and single-center design restrict broader applicability. We also recognize that our C-section rate of 65% is high compared to the national C-section delivery rate in 2023 of 32.4%, according to provisional CDC numbers. We figured out that the overall C-section rate for our hospital during the recruitment period does not reflect the overall rates, so there is some selection bias. Nevertheless, the study establishes a foundation for understanding OXT exposure at birth, which may be relevant to clinical settings with comparable demographics and practices. Future studies with larger, more diverse cohorts and multi-center participation must confirm these findings and expand the generalizability of maternal OXT administration effects.

## 5. Conclusions

Given our understanding of the significant role of OXT in extrauterine bonding between mothers and babies and its importance in parturition, our study offers compelling evidence that the placental homeostasis of OXT may be vital to fetal development. The intricate mechanism underlying the regulation of OXT in the placenta supports the hypothesis that OXT is involved in fetal organogenesis, particularly in the retina. While focusing on the effect of OXT, specifically in cultured RPE cells, we determined that genes responsible for cellular proliferation, migration, and invasion were inhibited. The activated genes belong to the class that protects RPE cell survival and function. Thus, the transfer of placental OXT to fetal circulation promotes cell survival and function during prenatal development. This research opens new avenues for exploring the physiological and pathological implications of OXT for placental function. These findings also identify potential therapeutic targets for disorders associated with placental dysfunction. Further investigations are warranted to elucidate the precise molecular pathways involved in the placental homeostasis of OXT and its implications for maternal and fetal well-being.

## Figures and Tables

**Figure 1 cells-13-01735-f001:**
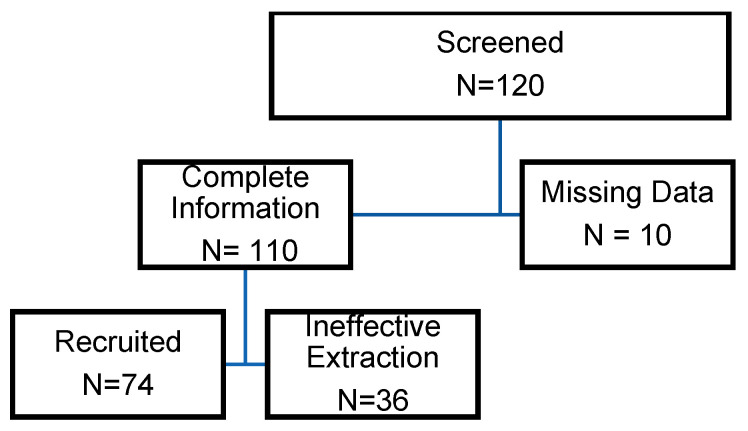
Infant enrollment and recruitment for the study.

**Figure 2 cells-13-01735-f002:**
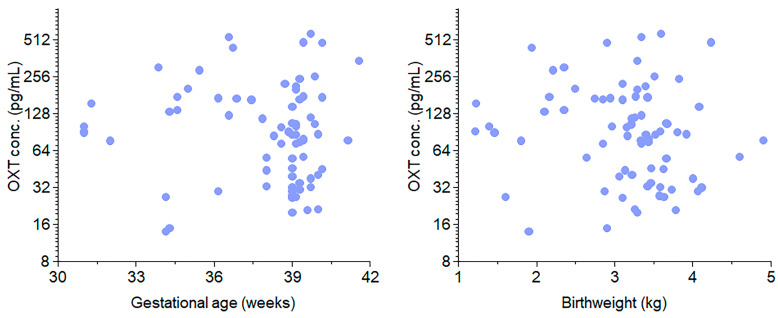
Distribution of OXT concentration in the cord blood by gestational age and birth weight. The graphs are scatter plots of OXT concentrations vs. gestational age (**left**) and OXT concentration vs. birthweight (**right**). In our study, term infants were predominately at 39 weeks with an average birth weight of 3.4 kg. No association was found between cord blood concentration of OXT and either birth weight or gestational age.

**Figure 3 cells-13-01735-f003:**
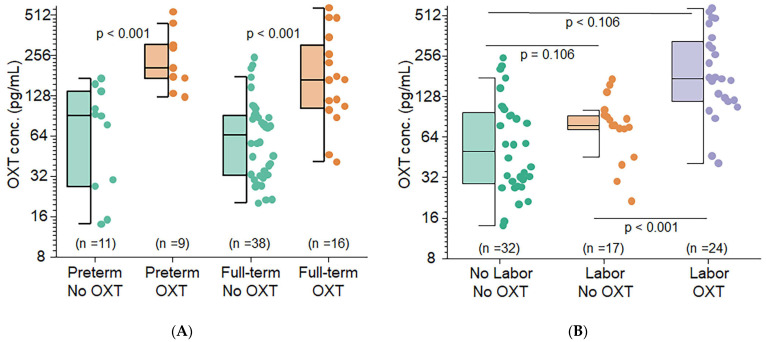
Maternal OXT crosses the placenta and affects fetal OXT concentration in preterm and term infants and during labor. (**A**) Concentration of OXT in the cord blood by preterm without (green) and with (orange) OXT. Similarly, a comparison of the concentration of OXT in the cord blood was made by full-term without (green) and with (orange) OXT. (**B**) Concentration of OXT in the cord blood by no labor without OXT (green), labor without OXT (orange), and labor with OXT (purple). Boxes represent the median, the bar within the box is the median, and the error bar represents the 1.5 interquartile range. Experimental data point distributions are all color-coded.

**Figure 4 cells-13-01735-f004:**
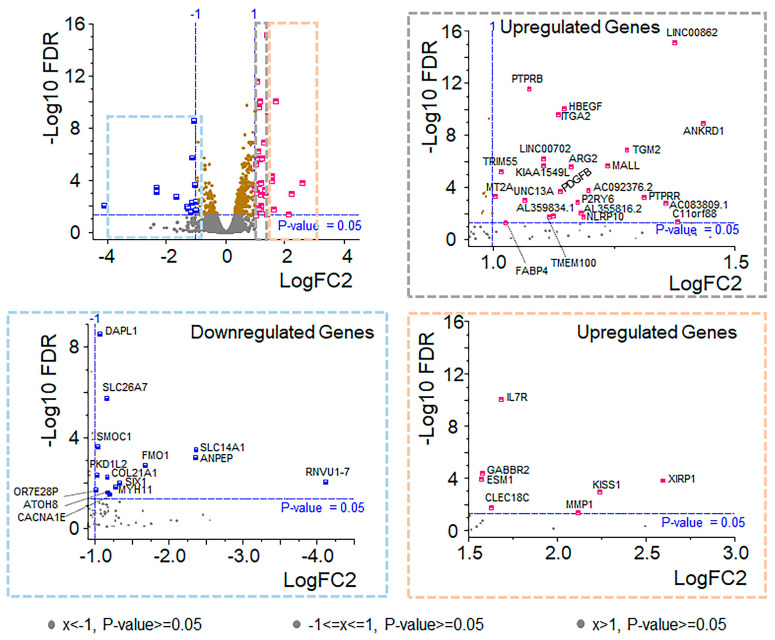
OXT regulated gene expression profile. Volcano plot of gene expression scatter as a function of statistical significance. Genes are colored if they pass the thresholds for FDR and log fold change (blue for downregulated and pink for upregulated). The top genes with a *p* value < 0.05 are labeled brown, and genes that did not meet our threshold are colored gray. Further segregation of downregulated (blue box) and upregulated (gray and orange boxes) genes are based on statistically significant fold changes.

**Figure 5 cells-13-01735-f005:**
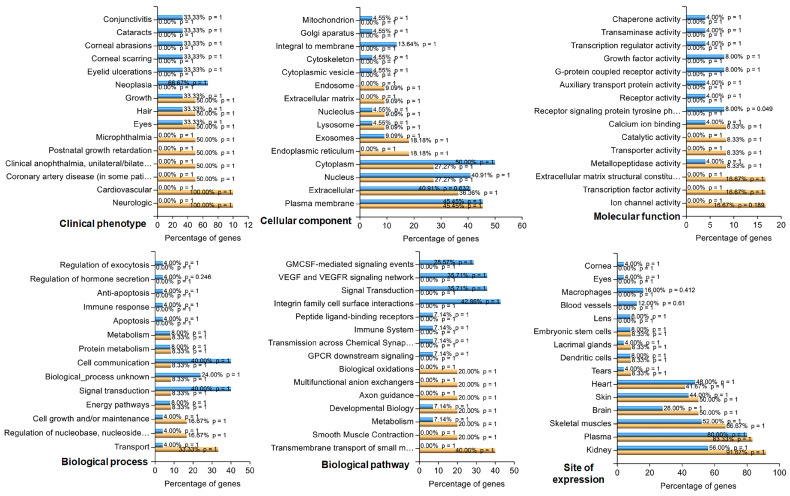
Top 15 crucial biological coverage based on analyzed transcript datasets. Functional enrichment of down- (orange) and upregulated (blue) genes following oxytocin treatment of human retinal pigment epithelial cells.

**Figure 6 cells-13-01735-f006:**
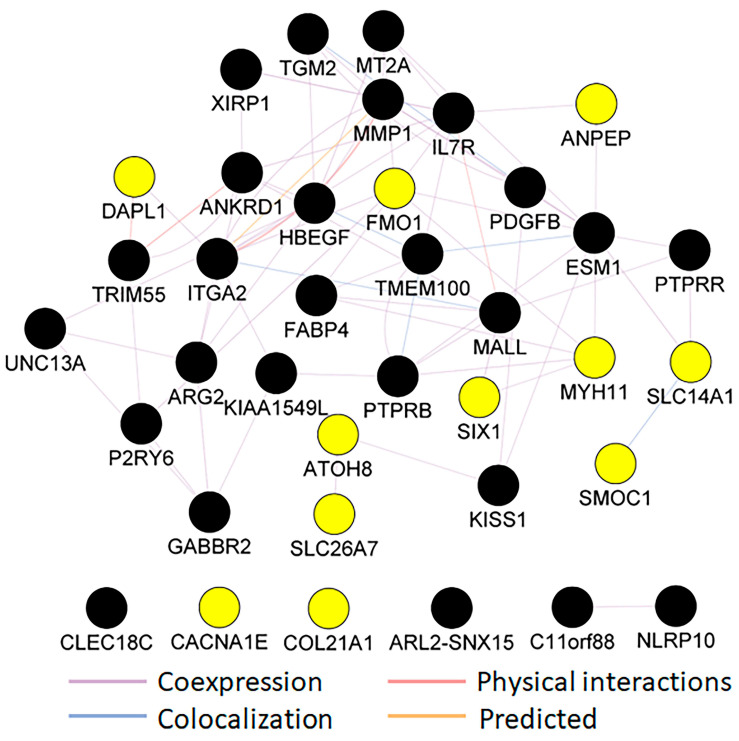
GeneMANIA connectivity mapping of DEGs based on threshold. Black nodes are upregulated genes, and yellow nodes are downregulated genes detected in this study.

**Figure 7 cells-13-01735-f007:**
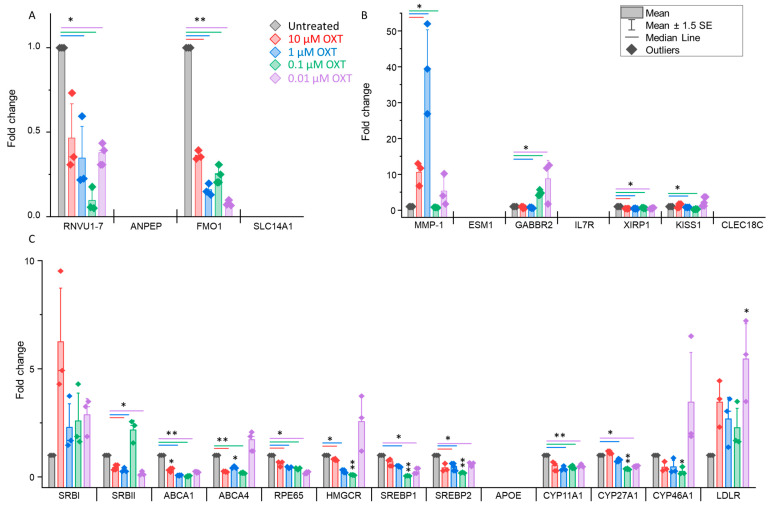
OXT treatment induces dynamic modifications in the expression profile of genes in hfRPE cells. (**A**) Expression profile of downregulated genes identified by RNA sequencing using real-time PCR. (**B**) Expression profile of some of the upregulated genes as identified by RNA sequencing using real-time PCR. (**C**) Expression profile of genes involved in cholesterol biosynthesis and metabolism. Fold change is presented as the mean ± SE. n = 3 biological replicates and *: *p* ≤ 0.05, **: *p* < 0.001 using student’s *t* test.

**Figure 8 cells-13-01735-f008:**
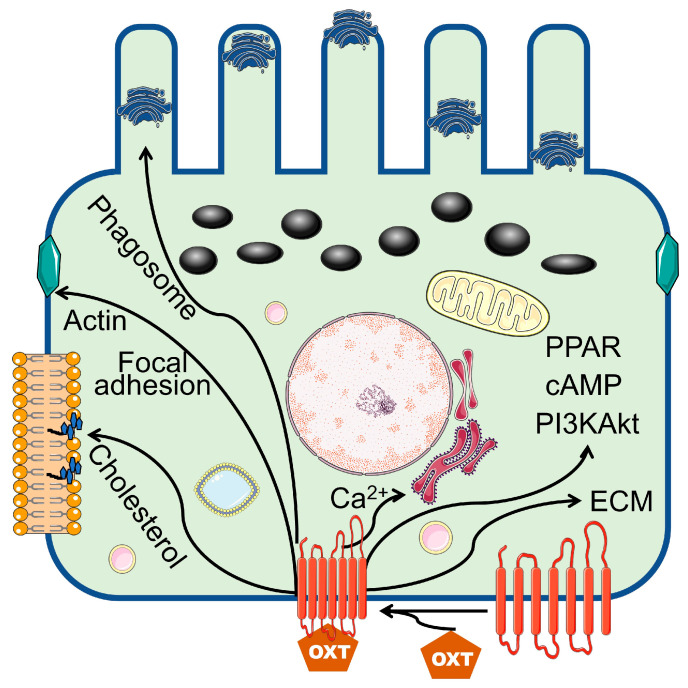
Summary representation of the OXT signaling pathway that might control cellular function. Illustration of the art borrowed from Servier Medical Art.

**Table 1 cells-13-01735-t001:** Maternal and infant perinatal and birth information, respectively.

	Preterm	Full-Term	
Characteristic	(n = 20)	(n = 54)	*p* Value
Gest. Age, Weeks			
Median [IQR]	34.5 [32.7–36.0]	39.2 [39.0–39.6]	
Birth Weight, kg			
Median [IQR]	2.19 [1.68–2.74]	3.43 [3.21–3.69]	
Labor, n (%)			0.039
No	5 (25)	28 (52)	
Yes	15 (75)	26 (48)	
Delivery, n (%)			0.246
C-Section	10 (50)	35 (65)	
Vaginal	10 (50)	19 (35)	
High Blood Pressure/			
Preeclampsia, n (%)			<0.001
No	8 (40)	48 (89)	
Yes	12 (60)	6 (11)	
Abruption, n (%)			0.070
No	18 (90)	54 (100)	
Yes	2 (10)	---	
Diabetes Mellitus, n (%)			
No	11 (55)	49 (91)	0.001
Yes	9 (45)	5 (9)	
Maternal Steroids, n (%)			<0.001
No	11 (55)	54 (100)	
Yes	9 (45)	---	
5-Min Apgar < 5			0.270
No	19 (95)	54 (100)	
Yes	1 (5)	---	
FiO_2_ > 21%			0.003
No	13 (65)	51 (94)	
Yes	7 (35)	3 (6)	
Maternal OXT, n (%)			0.214
No	11 (55)	38 (70)	
Yes	9 (45)	16 (30)	
Cord Blood [OXT], pg/mL			0.106
Median [IQR]	137 [71.1–223]	82.1 [40.3–146]	

Abbreviations: interquartile range (IQR), fraction of inspired oxygen (FiO_2_).

## Data Availability

Sequencing datasets will be deposited into Bioproject (https://www.ncbi.nlm.nih.gov/bioproject/) promptly following publication, and we will abide by the NIH Genomic Data Sharing Policy (https://gds.nih.gov/).

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
