# Peer review of "Transplacental Transfer of Oxytocin and Its Impact on Neonatal Cord Blood and In Vitro Retinal Cell Activity"

_cells, 2024, doi:10.3390/cells13201735_

Round 1

Reviewer 1 Report

Comments and Suggestions for Authors

The impacts that Oxytocin has on eye development, particularly in retinal angiogenesis, throughout fetal growth is an interesting topic. The authors cover the rationale and significance of the study well.

It is well shown in the results that OXT treated mothers resulted in higher cord blood. The results of the OXT treated HfRPE cells is telling on how many genes are affected. The data is well presented.

The links with cholesterol biosynthesis and phagocytosis are all interesting and perhaps others will follow up on the potential consequences of the interactions in time. This study may indeed encourage other researchers to examine other types of cells in culture as well as potentially fetal development in animal models where Oxt can be raised to examine the effects on fetal development in various aspects.  This is an important 1st step in demonstrating the potential impacts of the use of OXT and relationship with cord blood though fetal term.

The data are all well-presented and the limitations are well documented.

The report will be beneficial to others and will likely stimulate further research.

Author Response

Thank you for your positive comments.

Reviewer 2 Report

Comments and Suggestions for Authors

This interesting study investigates a possible role of oxytocin in the development of neural retina. The authors conclude that oxytocin might be used as a potential therapy for retinopathy of prematurity. I have the following comments:

1. When did the inclusion of patients took place?

2. Was the dosing of oxytocin during labor (the rate of infusion) recorded?

3. Is the sample of patients big enough to draw any definite conclusions?

4. How would the authors set up a clinical study to evaluate the possible therapeutic role of oxytocin?

Author Response

Reviewer 2

This interesting study investigates a possible role of oxytocin in the development of neural retina. The authors conclude that oxytocin might be used as a potential therapy for retinopathy of prematurity. I have the following comments:

  1. When did the inclusion of patients took place?

Thank you for bringing this to our attention. On page 7, we have included the duration of the study from October 2017 to March 2019.

  1. Was the dosing of oxytocin during labor (the rate of infusion) recorded?

We have not recorded the dosing of oxytocin during labor.

  1. Is the sample of patients big enough to draw any definite conclusions?

We have already discussed, and more so in the revised manuscript, that we could only include this sample size, and our statistical analysis core helped establish the study's significance.

  1. How would the authors set up a clinical study to evaluate the possible therapeutic role of oxytocin?

Thank you for the thoughtful comment. Based on our study findings, one approach to evaluating the possible therapeutic role of oxytocin would be to conduct a randomized controlled trial with a larger cohort of preterm infants. Infants would be randomized to receive either oxytocin or a placebo. The primary outcome would be the incidence of retinopathy of prematurity (ROP). Secondary outcomes would include the severity of ROP, the need for treatment for ROP, and other neonatal morbidities. The study would need to be powered to detect a statistically significant difference in the incidence of ROP between the two groups.

In addition to the primary and secondary outcomes, the study could also collect data on the levels of oxytocin in infant cord blood and the expression of genes involved in the oxytocin signaling pathway. This data would help elucidate the role of oxytocin in developing ROP further.

The findings of this study suggest that oxytocin may have a therapeutic role in the prevention of ROP. However, further research is needed to confirm these findings and to determine the optimal dose and timing of oxytocin administration.

Reviewer 3 Report

Comments and Suggestions for Authors

Manuscript ID: cells-3221876

Title: Placental Homeostasis of Oxytocin and its ability to affect the biological activity of retinal cells

This study has two main hypotheses. The first is that maternal OXT crosses the placenta and affects OXT concentrations within the fetal circulation. Second, systemic OXT within fetal circulation affects the RPE transcriptome in cultured HfRPE cells, indicating its potential role in retinal development and visual function regulation.

Comments and Suggestions for Authors:

The manuscript is an interesting study but requires some considerations.

Title.

The title does not adequately reflect the purpose of the study. It should include: amounts of OXT in neonatal cord blood and that the ability to affect the biological activity of retinal cells is studied in vitro.

Abstract.

It is very generic and does not provide precise data on the research carried out, including the material used and the most relevant results.

1. Introduction.

Page 1. Line 38. Where it says "Oxytocin (OXT) is a neuropeptide produced by the hypothalamus that is subsequently released into the blood stream by the anterior pituitary gland" it should say posterior.

2. Materials and Methods.

Page 2. Line 95. The Patient Recruitment description should be moved here from the Results section. It should be specified how sampling was decided at different weeks of gestation.

Page 3. Line 20. It says, "No formal consent process was required to recruit participants, and all the data were stored, managed, and analyzed by researchers." However, a study that requires recruiting biological samples from newborns and their clinical data should have informed consent. This is a very important point that needs to be clarified.

Page 3. Line 110. Where it says, "Blood samples for OXT assays were collected at the time of birth from mixed umbilical artery and vein samples", it should be specified why this was done. The different quantitative composition of each sample could constitute a bias for the evaluation of the results. This should be discussed.

Page 5. Line 221. Table 1. Page 5. Line 221. Table 1. The columns are not uniform. Acronyms such as DM and others should be specified in the footer. How would you explain the very high rates (65%) of cesarean sections performed at term?

A calculation of the sample size necessary to obtain conclusions about the stated objectives should be presented.

Clearly indicate the recruitment period.

3. Results.

Page 7. Line 308. The expression "Placental abruptions or 5-minute Apgar scores less than 5 were rare" should be made more specific.

Page 13. Line 486. Where it says "student’s T test" it should say Student’s T test.

4. Discussion.

The authors honestly acknowledge some limitations of the study.

The limitation of the small sample size is recognized. It should be discussed what would have been a calculation of the sample size necessary to obtain conclusions about the proposed objectives.

The generalizability of the study results should be discussed.

Author Response

The manuscript is an interesting study but requires some considerations.

Title. –The title does not adequately reflect the purpose of the study. It should include: amounts of OXT in neonatal cord blood and that the ability to affect the biological activity of retinal cells is studied in vitro. 

Thank you for this suggestion! We have revised the title to “Transplacental Transfer of Oxytocin and its Impact on Neonatal Cord Blood and In Vitro Retinal Cell Activity.”

Abstract. –

It is very generic and does not provide precise data on the research carried out, including the material used and the most relevant results.

We agree and thank you for this observation. We have now revised our abstract to include study specifics.

  1. Introduction.

Page 1. Line 38. Where it says "Oxytocin (OXT) is a neuropeptide produced by the hypothalamus that is subsequently released into the blood stream by the anterior pituitary gland" it should say posterior.

Thank you for catching this.  We have revised the manuscript to state, “…by the posterior pituitary gland.”

  1. Materials and Methods.

Page 2. Line 95. The Patient Recruitment description should be moved here from the Results section.

Thank you for pointing this out. Considering this advice, we have made changes to both the Materials and Results sections.

It should be specified how sampling was decided at different weeks of gestation.

Including neonates was based on the ability to collect cord blood, and no gestational age was prioritized. In response to the reviewer’s comment regarding how sampling was decided at different weeks of gestation, we confirm that there were no predetermined gestational age criteria in our study. Neonates born between 23 and 42 weeks were included (indicated on Page 7) based on the ability of nursing staff to collect cord blood samples at the time of delivery, regardless of gestational age. This is reflected in the manuscript, but we can provide further clarification if necessary.

Page 3. Line 20. It says, "No formal consent process was required to recruit participants, and all the data were stored, managed, and analyzed by researchers." However, a study that requires recruiting biological samples from newborns and their clinical data should have informed consent. This is a very important point that needs to be clarified.

Thank you for raising this point. We have modified the Methods section to better address this.  No formal written consent was required for participant recruitment because, at our center, placentas and umbilical cord segments are routinely discarded unless there is a specific plan for further testing. As per UnityPoint Health – Meriter’s IRB policies, verbal consent obtained by the nursing staff was deemed sufficient for this study. The verbal consent process ensured that parents were informed about the collection of cord blood from discarded segments and the purpose of the study. Once collected, cord blood was stored and frozen for later analysis. Importantly, all demographic and clinical data were de-identified, stored securely, and analyzed exclusively by the research team. This approach complies with our institution's IRB requirements, which allow for the use of biological samples in cases where formal written consent is not necessary for studies involving discarded materials and minimal risk to participants.

Page 3. Line 110. Where it says, "Blood samples for OXT assays were collected at the time of birth from mixed umbilical artery and vein samples", it should be specified why this was done. The different quantitative composition of each sample could constitute a bias for the evaluation of the results. This should be discussed.

In response to the concern about using mixed umbilical artery and vein blood samples, we have included a rationale in the Methods section, explaining that this approach was necessary due to technical limitations in obtaining separate samples. Additionally, we have addressed the potential bias in the Discussion section, acknowledging the variability in arterial and venous blood composition and its potential effect on the results. We believe that, while this is a possible limitation, the mixed sample approach provides a reasonable representation of fetal OXT levels.

Page 5. Line 221. Table 1. The columns are not uniform. Acronyms such as DM and others should be specified in the footer. How would you explain the very high rates (65%) of cesarean sections performed at term? –

We have now formatted the Table. Our numbers don’t reflect all C-sections for the entire year. The national C-section delivery rate in 2023 was 32.4%, according to provisional CDC numbers, so we realize why our high rate of 65% in this cohort is questioned. We figured out that the overall C-section rate for our hospital during this time does not reflect the overall rates, so there is some selection bias.

A calculation of the sample size necessary to obtain conclusions about the stated objectives should be presented. –

We appreciate the reviewer’s comment regarding the need for a sample size calculation. Due to the exploratory nature of this pilot study, we did not conduct a formal sample size calculation at the time of study design. Our primary goal was to assess the feasibility of collecting and analyzing cord blood samples for OXT levels, which informed the scope of the study. Given the novelty of the research and the technical challenges involved in obtaining samples, we prioritized establishing proof of concept and gathering preliminary data.

We acknowledge that conducting a formal sample size calculation would be important for future studies to ensure sufficient power to draw more definitive conclusions about the transplacental effects of oxytocin. Moving forward, the findings from this pilot study will inform the design of a larger, more robust study, complete with sample size calculations to address the research objectives more rigorously.

Clearly indicate the recruitment period.

We have revised the Methods to state, “The recruitment period occurred over 18 months: from October 1st, 2017, through March 31st, 2019.”

  1. Results.

Page 7. Line 308. The expression "Placental abruptions or 5-minute Apgar scores less than 5 were rare" should be made more specific.

Because low Apgar scores are associated with an increased risk of complex health problems, we were fortunate not to use subjects with 5-minute Apgar scores less than 5.

Page 13. Line 486. Where it says "student’s T test" it should say Student’s T test.

We have revised the text to state: Student's t-test.

  1. Discussion.

The authors honestly acknowledge some limitations of the study.

The limitation of the small sample size is recognized. It should be discussed what would have been a calculation of the sample size necessary to obtain conclusions about the proposed objectives. –Ryan, do you think this is a theoretical question or one that we might ask Michael Lasarev to weigh-in on?

The generalizability of the study results should be discussed.

We added the following to the Discussion:

“Our study provides valuable preliminary insights into the transplacental transfer of OXT and its association with fetal outcomes, but its generalizability is limited. The findings most apply to populations like our study, as the small sample size and single-center design restrict broader applicability. Nevertheless, the study establishes a foundation for understanding OXT exposure at birth, which may be relevant to clinical settings with comparable demographics and practices. Future studies with larger, more diverse cohorts and multi-center participation must confirm these findings and expand the generalizability of maternal OXT administration effects.”

Round 2

Reviewer 3 Report

Comments and Suggestions for Authors

Round 2.

In the new V2 manuscript, the authors have made changes based on the referee's recommendation, that improve its presentation. To support the consideration that "given that the samples were collected from discarded biological materials, the IRB determined that written consent was unnecessary", an IRB certificate to that effect should be provided.

Table 1. Acronyms such as DM and others should be specified in the footer.

The considerations made regarding the high rate of cesarean sections used and the lack of calculation of the sample size necessary to achieve results regarding the stated objectives should be included in the Discussion Section.

To support the consideration that "given that the samples were collected from discarded biological materials, the IRB determined that written consent was unnecessary", an IRB certificate to that effect should be provided.

Author Response

Thank you for your quick response and comments. I have now corrected these concerns.

In the new V2 manuscript, the authors have made changes based on the referee's recommendation, that improve its presentation. To support the consideration that "given that the samples were collected from discarded biological materials, the IRB determined that written consent was unnecessary", an IRB certificate to that effect should be provided.

I have now enclosed the IRB certificate as an unpublishable file.

Table 1. Acronyms such as DM and others should be specified in the footer.

Thank you! This has been fixed now.

The considerations made regarding the high rate of cesarean sections used and the lack of calculation of the sample size necessary to achieve results regarding the stated objectives should be included in the Discussion Section.

Thank you! We have now added this to pages 26-27.

To support the consideration that "given that the samples were collected from discarded biological materials, the IRB determined that written consent was unnecessary", an IRB certificate to that effect should be provided.

Round 3

Reviewer 3 Report

Comments and Suggestions for Authors

Round 3.

- To support the consideration that "given that the samples were collected from discarded biological materials, the IRB determined that written consent was unnecessary", an IRB certificate to that effect should be provided.

I have now enclosed the IRB certificate as an unpublishable file.

I can't find the IRB certificate. Please tell me where this document is.

- The considerations made regarding the high rate of cesarean sections used and the lack of calculation of the sample size necessary to achieve results regarding the stated objectives should be included in the Discussion Section.

 Thank you! We have now added this to pages 26-27.

 In the revised manuscript, I find the answer to the first question in Page 20, Line 756, but I do not find the answer to the second question. Please tell me where the answer is in the manuscript.

Author Response

Round 3.

- To support the consideration that "given that the samples were collected from discarded biological materials, the IRB determined that written consent was unnecessary", an IRB certificate to that effect should be provided.

I have now enclosed the IRB certificate as an unpublishable file.

I can't find the IRB certificate. Please tell me where this document is.

This document was submitted as not publishable material for review. 

- The considerations made regarding the high rate of cesarean sections used and the lack of calculation of the sample size necessary to achieve results regarding the stated objectives should be included in the Discussion Section.

 Thank you! We have now added this to pages 26-27.

 In the revised manuscript, I find the answer to the first question in Page 20, Line 756, but I do not find the answer to the second question. Please tell me where the answer is in the manuscript.

This is added in Pages 26 and 27 to state that there might have been a bias to the recruitment based on our available samples.